# Schools That 'Open Doors' to Prevent Child Abuse in Confinement by COVID-19

**Esther Roca** [1],*[ID]**, Patricia Melgar** [2][ID]**, Regina Gairal-Casadó** [3][ID]
**and Miguel A. Pulido-Rodríguez** [4][ID]

1   Department of Comparative Education and Education History, University of Valencia, Av. Blasco Ibáñez 30, 46010 València, Spain
2   Department of Pedagogy, University of Girona, Pl. Sant Domènec, 9, 17004 Girona, Spain; patricia.melgar@udg.edu
3   Department of Pedagogy, Rovira i Virgili University, Ctra. de Valls, s/n, 43007 Tarragona, Spain; regina.gairal@urv.cat
4   School of Social Education and Social Work Pere Tarres, Ramon Llull University, Carrer Santaló, 37, 08021 Barcelona, Spain; mapulido@peretarres.url.edu
*   Correspondence: esther.roca@uv.es

**Abstract:** Background: Due to the expected increase in child abuse during the period of COVID-19 confinement, it is essential that social researchers and other professionals work together very quickly to provide alternatives that protect children. To respond to this extremely urgent demand, evidence-based actions are presented that are being carried out in nine schools in the autonomous communities of Valencia and Murcia, Spain, during the confinement with the goal of "opening doors" to foster supportive relationships and a safe environment to prevent child abuse. Methods: The research was conducted through the inclusion of teachers who are implementing these actions in dialogue with the researchers to define the study design, analysis, and discussion of the results. Results: Knowledge regarding six evidence-based actions is provided: (1) dialogic workspaces, (2) dialogic gatherings, (3) class assemblies, (4) dialogic pedagogical gatherings with teachers, (5) mixed committees, and (6) dynamisation of social networks with preventive messages and the creation of a sense of community, which are being implemented virtually.

**Keywords:** child abuse; COVID-19; prevention

## 1. Introduction

According to the UNESCO Institute for Statistics, more than 188 countries closed their educational centres at all levels during the COVID-19 pandemic as one of the measures to stop the spread of the virus. More than 1,500,000,000 learners were affected, which is more than 90% of all enrolled learners, and were confined in their homes [1]. In Spain, all educational centres from early childhood education to universities were closed on Thursday 13 March 2020. The next day, a state of alarm was decreed in the country. Citizens were asked to remain confined to their homes.

Faced with this global situation, on 20 March 2020, the Center on the Developing Child at Harvard University highlighted the imperative need to reconcile two of the most relevant science-based messages for overcoming the COVID-19 pandemic, prolonged social distance and supportive relationships, to strengthen resilience in the face of adversity. The Center noted that responding from science to two apparently contradictory challenges requires rigorous scientific thinking, on-the-ground expertise and the lived experiences of a wide range of people and communities. Obviously, the impact of confinement conditions will not have the same short- and long-term effects on all people [2]. It is a priority for

social and educational research to contribute to the challenge of reducing the potential threats and consequences that the physical isolation of confinement can produce for those who are especially vulnerable, such as children at risk of abuse in their homes. This article provides knowledge to help minimise the negative consequences of confinement for all children, especially for those potentially at risk of abuse (physical, sexual, and emotional) at home.

In response to this challenge, this research provides knowledge from six evidence-based actions that are being carried out in nine schools (pre-primary, primary, middle, high, and special education) in the autonomous communities of Valencia and Murcia, Spain, during the period of COVID-19 confinement with the goal of "opening doors" to foster supportive relationships and a safe environment to prevent child abuse. The research was conducted with the inclusion of teachers who are implementing these actions in dialogue with the researchers to define the study design, analysis, and discussion of the results.

## 1.1. Increased Domestic Violence and Child Abuse during Confinement

On March 24th, Marceline Naudi, a representative of the Group of Experts on Action against Violence against Women and Domestic Violence (GREVIO), the organisation within the Council of Europe responsible for the supervision of the Istanbul Convention [3], made a call to create alternatives for women and their children who suffer domestic violence in the face of the COVID-19 crisis and the preventive measures of confinement. Due to the expected increase in domestic violence, it is essential for researchers and other professionals to work together and share actions to protect not only women but also their children [4].

Research estimates that 275 million children worldwide are exposed to domestic violence, including physical, sexual, and emotional abuse, which may also involve neglect or deprivation [5]. UNICEF cautions that this is a conservative estimate based on the limitations of available data. In fact, millions more children may be affected by domestic violence. Violence at home is not limited by geography, ethnicity or status; it is a global phenomenon. Confinement due to the pandemic forces these children to live with their abusers all the time. Child abuse is one of the main causes of emotional, behavioural and health problems throughout life [6].

According to the Ministry of Equality of the Spanish Government, during the first fortnight of April, there was an exponential increase of 48% of phone calls to the 016 telephone number addressed to assist victims of violence against women and their children, as well as an increase of 650% in online consultations with the different existing services [7]. In Spain, non-governmental organizations have warned that many children and adolescents are suffering from increasing violence and lack of protection during confinement. The only available data are from the ANAR Foundation (Help Children and Adolescents at Risk, in its Spanish acronym). This foundation has chosen to replace its usual telephone assistance service by a chat due to the possible difficult circumstances of calling in which boys and girls who need help can find themselves. The data collected from the chat show an increase of 300% for queries at this time. During confinement, it has received an average of 38 daily consultations [8], while during 2019 the daily average was 9 consultations [9]. In only one week (23–30 March), the professional team of this organisation answered 270 requests for assistance that were received by chat and e-mail. Among these, 173 serious cases were detected. Two out of every five cases addressed were due to violence. Physical violence within the family constituted the highest percentage at 12.7% of cases, followed by psychological abuse (6.9%). Noteworthy for their seriousness were cases of sexual abuse (3.5%), in which the minors could not escape from their aggressors. A total of 2.9% were minors who were in a state of abandonment or neglect during this period. Requests for help were made by children who had access and skills to use electronic devices through the Internet. ANAR warns that these data are only a small part of what is truly happening. Younger children are even more unprotected if neighbours, friends, and family do not identify and report situations that may be occurring [10]. Research indicates that "closed doors", hiding, non-identification, and non-reporting are the main barriers to overcoming child abuse [11].

### 1.2. Keeping Supportive Relationships Alive through Schools during Confinement

Other evidence-based messages from the scientific community in education are that teachers should continue to maintain positive and stable relationships with their students and actively promote ways to create a sense of community during the period of confinement [12]. Expanding a school's capacity for support and ensuring that each student is known to at least one adult could help to identify students who may be at risk during times of increased isolation [13] and can also have enormous benefits, including less bullying, lower dropout rates, and improved social and emotional capacities [14].

### 1.3. School-Based Programs to Prevent Child Abuse

In the last three decades, evidence has been collected on school-based abuse prevention programmes from early childhood, elementary and high schools; these programmes have been found to be effective in increasing children's knowledge of child abuse concepts and self-protection skills [6,15,16]. Schools are ideal settings for promoting child abuse prevention, providing knowledge and skills for children to recognise abuse and to reduce risk, creating a trusted environment in which children can disclose if they are being abused, and creating a safe context involving the entire community [6,15,17–20]. The main messages in programmes that have proven to be effective are to tell a trusted adult, that it is never too late to tell, that the abuse is never their fault, and that the perpetrator is always responsible. Examples are provided to further define these concepts so that children understand what is meant by "trusted adult" and "unsafe or confusing touch" [17], permission to say no to authority figures, understanding that trusted adults can act abusively, and rules about breaking promises and keeping secrets [20]. The purpose of these programmes is to transfer the knowledge and skills learned by children or adolescents in classroom settings to real-life situations [21].

Other programmes focused on promoting healthy relationships and preventing domestic abuse emphasise friendships and peer relationships to discuss abuse in intimate relationships, how to build positive relationships, how children and youth can develop relationships free of fear and abuse, conflict resolution skills, and underlying attitudes that lead to abusive tendencies [22]. In the same vein, evidence-based interventions following a preventive socialisation of violence approach have been conducted. Research on risk factors related to gender violence conducted from a preventive socialisation approach has identified the existence of a coercive dominant discourse in which people with violent attitudes and behaviours are socially portrayed as attractive and exciting, while people and relationships with nonviolent attitudes and behaviours are portrayed as less exciting [23,24]. Some of the messages from this approach are directed towards the promotion of zero violence from early childhood education, such as "a friend is the one who treats you well" and "someone who is an upstander in favour of the victim, acts without violence, and reports aggression is not a snitch, he/she is brave". These messages aim to ensure that there is no justification for violence from an early age and to promote the socialisation based on the desire of nonviolent people who have the best values [25]. In this approach, the actions addressed towards the preventive socialisation of violence are not separated from instrumental learning sessions (mathematics, language, science, etc.) or from the different spaces of the school (class, gym, lunchroom, etc.) Instead, they are promoted in each of the interactions that occur in all of these spaces [26,27].

These interventions work by implementing effective pedagogical principles used by teachers and programme facilitators in the classroom, especially those based on sociocultural theories of learning such as Vygotsky's theory [21]. From this perspective, evidence has shown that the most effective programmes are those that provide opportunities for active participation, including role-playing, video modelling, and discussion [6]. Along these lines and from the perspective of the preventive socialisation of violence, schools are developing dialogic gatherings with boys and girls to promote egalitarian dialogues focused on the transformation of the language of desire to create possibilities for those who wish to question desires imposed by patriarchal societies and orient these desires towards nonviolent relationships [28]. Many of these interventions are designed to engage children in interactive activities

in which they exchange viewpoints on these issues [29,30]. For example, in interventions focusing on the prevention of sexual abuse, they discuss situations in which an adult's touches or kisses feel like a boundary violation and how to deal with such situations [31]. Interventions aimed at adolescents also encourage interaction among themselves and participation, largely by using real-life stories to engage students in discussion, identify with the different actors in a story, share their opinions, and listen to the opinions of others [22].

Research has also been conducted on how these effective school-based programmes to prevent child abuse train teachers [32,33]. Training that targets teachers aims to raise awareness of their potential to be agents of social change to promote detection, disclosure, and intervention in their daily work at school [34,35]. In these trainings, teachers are warned of the risk children face when they disclose abuse and are not believed or are blamed, which can have devastating and long-lasting psychological, physical, relational, educational, and social effects. Therefore, teachers are encouraged to be agents of change who contribute to the creation of a safe environment to optimise the likelihood that children feel safe in disclosing child abuse, and that when children do disclose, they are believed and supported in the process [17]. Another effective action in teacher training are dialogic pedagogical gatherings [36]. In these gatherings, scientific articles and books with the main social, educational and psychological contributions related to child abuse and violence prevention are read and discussed by teachers [26].

Other research has highlighted as a success factor the inclusion of local elements that promote community involvement in designing and developing child abuse prevention programmes [19]. Research on child abuse prevention is clear that equipping children with greater protective skills and knowledge does not replace the responsibility of society to ensure the safety of children [15]. For this reason, some of these child abuse prevention trainings also target all school staff, families, and community members [6], enabling them to identify inappropriate situations and react appropriately by responding quickly and effectively to disclosures to protect children from new abuse, thus creating a safe context to prevent child abuse [31]. In this regard, some interventions have been aimed at promoting upstanding behaviour involving children, families, teachers, and other school staff by being an upstander who, directly or indirectly, "says no" to violence, thus overcoming the role of the passive bystander who, knowingly or not, colludes with and supports abusive behaviour [37–39]. Other interventions that involve the community start from a dialogic model of conflict prevention and resolution in which students, teachers, and families are involved in the decision making of norms for the promotion of a safe environment that is free of violence [27].

Transferring these evidence-based messages into actions that can be promoted by schools in times of confinement could be essential to reduce the risk of child abuse. This study does not aim to identify children at risk of abuse. Since child abuse has been considered a kind of aggression that takes place "behind closed doors" [11], the systematization of ODA offers knowledge to replicate them. The dissemination of ODA could contribute to "open doors" in other confined homes, creating spaces for supportive interactions as a preventive factor of child abuse and for good social, emotional, and physical development [40]. For the abovementioned references, the aim of this work is to provide knowledge on evidence-based actions to "open doors" to foster supportive relationships and a safe environment to prevent child abuse during the period of COVID-19 confinement outside of schools. In a co-creation process between teacher leaders and researchers, what has been called the Open Doors Actions (ODA) emerge, based on scientific evidence with social impact for the creation of relationships that support children in times of physical distancing [41]. These actions are based on the transfer of Successful Educational Actions [42] to virtual school-home spaces. The article presents a systematization of these actions through a first analysis after six weeks of implementation. In this sense, the article does not show results resulting from the implementation of said actions. The impact of these actions is the subject of further studies and articles that are currently in progress. Therefore, it is a priority for the actions that are presented here—such as the dialogic workspaces, the dialogic gatherings, the class assemblies, the dialogic pedagogical gatherings with teachers, the mixed committees and the

dynamisation of social networks with preventive messages and the creation of a sense of community—to be available to the worldwide educational community.

## 2. Materials and Methods

This study was developed following a communicative methodology [43–45]. In the framework of this communicative approach, Dialogic Recreation of Knowledge (DRK) [25] has been implemented. The main characteristic of DRK is the equal basis on which the dialogue between researchers and participants takes place. This process is conducted on the basis of the validity of the arguments in contrast with the scientific evidence. Principals and teachers from participating schools and researchers establish a dialogue and jointly analyse the actions that can enable the challenge of "open doors" to promote supportive relationships and safe environments for child abuse prevention during COVID-19 confinement.

Due to the situation of confinement, fieldwork was developed in an online form between 18 March and 10 April 2020. Three sessions of a communicative focus group (CFG) were conducted in addition to five interviews with principals, teachers, and a school counsellor from nine different schools (pre-primary and primary, secondary and special education). The creation of a WhatsApp group with the members of the CFG and a collaborative Excel sheet shared through Drive was also started with a register of the actions that the teachers entered. Table 1 shows this information (CFG, WhatsApp group -WP-, Interview -Int-). To assure anonymity, each participant and school was assigned a code. For participants, the first letter corresponds to their educational profile, T (Teacher), P (Principal), H (Head teacher), or C (school Counsellor), and the second letter corresponds to P (Primary), S (Secondary), or E (Special Education). Finally, a correlative number is indicated. For schools, the codification is similar: PS (Primary School), SS (Secondary School), and ES (Special Education School), adding a correlative number for the schools with the same profile.

**Table 1.** Participants' profiles in each data collection technique.

| Profile | Gender | Age | Years in School | School | Public/Private | CFG1 | CFG2 | CFG3 | WP | Int |
|---------|--------|-----|-----------------|--------|----------------|------|------|------|----|----|
| PP1 | Female | 41–50 | 10 | PS1 | Public | X | X | X | X | X |
| TP1 | Male | 41–50 | 5 | PS2 | Public | X | X | X | X | |
| PP2 | Female | 51–60 | 16 | PS3 | Public | X | X | X | X | |
| TP2 | Female | 41–50 | 3 | PS4 | Public | X | X | | X | X |
| TP3 | Male | 31–40 | 2 | PS5 | Public | X | X | X | X | |
| HP1 | Female | 31–40 | 13 | PS6 | Public | X | X | X | X | |
| TP4 | Female | 31–40 | 2 | PS2 | Public | | | | X | X |
| TS1 | Female | 41–50 | 20 | SS1 | Private | X | X | X | X | |
| CS1 | Female | 41–50 | 13 | SS2 | Public | X | X | | X | X |
| PE1 | Female | 41–50 | 20 | ES3 | Public | X | X | X | X | X |

Prior to data collection, the participants were informed about the aim of the study, that their participation was anonymous and voluntary, and that the data would be treated confidentially and would only be used for research purposes. All participants agreed to provide researchers with relevant data for the purpose of the study. Then they all signed an informed consent. The study respects the ethical guidelines of the European Commission (Ethics Review of the European Commission. FP7, 2013) and was approved by the Ethics Board of the Community of Researchers in Excellence for All (CREA) The Ethics Board was composed by: Dr. Marta Soler (President), who has expertise in the evaluation of projects from the European Framework Programme of Research of the European Union, and of European projects in the area of ethics; Dr. Teresa Sordé, with expertise in the evaluation of projects from the European Framework Programme of Research and research in the area of Roma studies; Dr. Patricia Melgar, founding member of the Catalan Platform Against Gender Violence, and researcher in the area of gender-based violence; Dr. Sandra Racionero, former secretary and member of the Ethics Board at Loyola University Andalusia (2016–2018), and review panel member for COST action proposals in the area of health; Dr. Cristina Pulido, expert in data protection policies

and child protection in research and communication; Dr. Oriol Rios, founding member of the "Men in Dialogue" association, researcher in the area of masculinities, as well as editor of *Masculinities and Social Change*, an indexed journal in WoS and Scopus; and Dr. Esther Oliver, who has expertise in the evaluation of projects from the European Framework Programme of Research and is a researcher in the area of gender-based violence.

*2.1. School Selection Criteria*

The selected schools are part of a network of schools in Europe and Latin America that implement educational actions with a social impact for the prevention of violence. The schools that participated in the study are also part of a subnetwork in the neighbouring territories of the regions of Valencia and Murcia, Spain. Given the urgency of producing the research results so that they would be available for use by other schools worldwide during the confinement period, the schools were selected through convenience sampling [46]. Schools were selected that were more easily accessible due to a close relationship with the researchers through training processes for the prevention of child abuse and violence. Among the nine selected schools, seven are pre-schools and primary schools. The other two are secondary and higher education schools. Eight of the schools are public and only one is a private school. There is a great diversity among the communities. Taking the socio-economic situation of the families as a reference, in three of the schools, more than 30% are at risk of social exclusion. In one of the schools, 100% of the students have special educational needs. In total, 2648 students have participated in ODA. All the teachers selected for the study are those who have led the implementation of ODA in their schools since the beginning of the confinement.

*2.2. Data Collection and Analysis*

2.2.1. Communicative Focus Groups (CFGs)

Three sessions were conducted with the same communicative focus group (CFG). Ten members of the school staff from participating schools took part. Notes were taken in written form during the first and second sessions of the CFG, and the third session was recorded and transcribed.

The goal of the CFG is to develop a dialogue and shared analysis of the situation after the closing of the schools as well as the possibility of implementing action to promote supportive relationships and safe environments to prevent child abuse.

On the 18th of March, the first CFG session was conducted. Participants first discussed the challenges and difficulties of reorganising distance learning, as well as making sure that all families have access to essential resources, making sure not to increase inequalities due to the confinement. Afterwards, dialogue was introduced about the evidence-based messages that oriented actions for the prevention of child abuse from the school and the community, such as (1) not hiding existing violence and increasing children's access to trust relationships that break the isolation; (2) promoting friendships and healthy relationships that protect children from toxic relationships; (3) ensuring supportive relationships to strengthen resilience in situations of adversity; (4) not revictimising children exposed to violence; (5) opening up spaces of dialogue to address issues about child abuse and violence prevention in the community; (6) including the voices of the children in the processes of conflict prevention and resolution; and (7) promoting upstander communities with an active stance against violence [6,15,17,22,24,25,27,28,31,39,47,48].

The second CFG session was conducted on 20 March. The discussion continued about ways to make the scientific criteria effective through actions. Some actions that were already being implemented during the confinement were exchanged, and new ones were defined. The third CFG session on 30 March assessed the first week of the implementation of actions and reflected on how to continue working with the schools to maintain the criteria in the actions that were developed online during the confinement period. Another topic that emerged in the focus groups was ways in which the COVID-19

infodemic should be addressed by the schools [49,50]. The actions that are being developed in these schools on this specific topic are the subject of other studies and articles in progress.

### 2.2.2. Interviews with a Communicative Approach

Five interviews were conducted (1 principal and 2 teachers of pre-primary and primary schools, 1 counsellor of a secondary school and 1 principal of a special education school). The goal of the interviews was to analyse how some of the actions that were introduced in their schools were being organised. One of the main topics discussed in the interviews was the implementation of the online dialogic literary gatherings and how they helped teachers create supportive relationships with students.

The interviews were used to obtain detailed information on how to recreate ODAs in non-classroom educational contexts. In the case of CS1, the interview was conducted because he could not be present for the entire duration of the second CFG. In the case of PP1, TP2, TP4, and PE1, the interviews were carried out on one of the ODAs in particular to elaborate on more details on the implementation of the action, which had not emerged in the dialogue of the CFGs.

### 2.2.3. CFG WhatsApp Group

A WhatsApp group was created to integrate the participants in the research. On the one hand, this group has an organising function for the exchange and collection of information. On the other hand, it promotes daily discussion about the implementation and introduction of the actions in the different schools. The research team shared evidence-based information that facilitated the implementation of the actions and the resolution of doubts. With the permission of the participants in the group, the communications were collected and used to complement the data of the study. An average of 33.5 interactions per day occurred during the period of development of the fieldwork.

### 2.2.4. Table of Collaborative Register

An Excel spreadsheet was shared in Drive with all the members of the CFG. The goal was for the members to provide information on each of the actions that they implemented or planned to implement in the short term in their schools. In particular, information was collected about the name of the action, its main objective, participants, frequency, timing, and some impacts detected in this first phase of implementation.

## 3. Results

The actions that schools implemented to promote supportive relationships and safe environments to prevent child abuse during the COVID-19 confinement were dialogic workspaces (DW) with students, teachers and volunteers, dialogic gatherings (DG) with students, class assemblies and mentoring (CA), dialogic pedagogical gatherings (DPG) with teachers and the community, mixed committees (MC) with teachers' families and other community members, and dynamisation of social networks with preventive messages and the creation of a sense of community (SN). In Table 2, the actions that were implemented in each school are presented. Each of the actions will be explained.

**Table 2.** Actions implemented by schools.

|     | DW | DG | CA | DPG | MC | SN |
|-----|----|----|----|-----|----|----|
| PS1 | X  | X  | X  | X   | X  | X  |
| PS2 | X  | X  |    |     | X  | X  |
| PS3 |    |    |    |     | X  | X  |
| PS4 |    |    |    |     |    | X  |
| PS5 |    | X  |    |     |    | X  |
| PS6 | X  | X  | X  |     | X  | X  |
| SS1 |    | X  | X  |     |    | X  |
| SS2 | X  | X  | X  |     | X  | X  |
| ES1 |    | X  | X  | X   |    | X  |

### 3.1. Dialogic Workspaces with Students, Teachers and Volunteers

Online DW are small groups of students connected through video conferences to work at home together with an adult, a teacher or a volunteer. In these schools, volunteers are included in the learning spaces [30]. The adult plays a fundamental role. On the one hand, the adults facilitate the interactions among students to encourage them to solve tasks together. On the other hand, they ensure that the interactions are respectful and of quality. The DW "open doors" turns what would normally be a private space where each child does homework individually at home into an open space with classmates and an adult person of reference who helps the group work together in an environment of support and trust. The principal of one of the pre-primary and primary schools explains it this way:

> "The online dialogic workspaces are a moment of connection between classmates in which other people participate, the teacher maintains quality interactions that allow them to continue learning together with their friends, having a dialogic space that gives purpose, motivates and cheers them up. It allows us to open up a public space in the homes". (PP1)

Online DW goes beyond enabling all children to successfully complete their homework. These workspaces also become a dialogic space where children can share their concerns and daily experiences. In the PS1 school, they start with a brief assembly about how the children are feeling at home so that they can express their concerns. If one of the children does not connect, the teacher calls the family to identify whether there is any problem that has prevented the connection. The director of the school explains how, despite the physical distance, the feeling is transferred to each family and each child that they are not alone and that teachers are concerned about the children's learning progress as well as their mood:

> "It is made clear to families and children that we do not put children aside. In this way, we can also see them and notice how they are doing, their mood, their expression. We continue fostering the learning process together with emotional support". (PP1)

In these spaces of joint work, some of the work that schools do involves positioning as an upstander against abusive behaviour. In one of the group sessions in that same school, one of the girls who had previously intimidated the same classmate had an attitude of rejection towards him, eliminating him from the online meeting. The rest of the classmates immediately acted as upstanders who did not allow that behaviour and informed the teacher about it. The principal, who also participated in online DW, highlights the importance of socialising in this joint position in the face of a violent attitude and how this can enable the children to transfer this to other spaces of their lives:

> "Something happened to us today in 4th of primary. A girl that usually annoys another [boy] in the class eliminated him from the videoconference. The rest reported it immediately, and I was able to intervene. We recalled the principles that guide our relationships, we remembered that we will not allow anyone to ill-treat anybody else. It is very important because it might be that in other spaces where they connect, without us there, they reproduce the rejection in these behaviours". (PP1)

### 3.2. Dialogic Gatherings

In dialogic gatherings (DGs), a collective construction of meaning and knowledge through the dialogue of the students is based on the best creations of humanity in different domains, such as literature, science, music, and other arts [29]. DGs were already implemented in participating schools before confinement, and now they are being developed online through a platform for video conferences, as are the DW.

The DG sessions are normally one hour a week. In each session, all students present their interpretation of what is being worked on in DG (a literary work, a piece of art, a musical composition, a painting, etc.). In the interventions, they relate what the work has stirred up in them in relation to the

issues of their concern. After each intervention, the teacher moderates an open debate, ensuring the equal participation of everyone. The principles of egalitarian dialogue and solidarity that ground the DG are aimed at ensuring that all interventions are respectful. The goal is to create a trust environment in which the children feel that they can share their experiences and reflections, as well as their feelings and emotions, and receive support from the interventions of their classmates. The teacher of the PS5 school explains this as follows:

> "We talk about very important issues that the children are concerned with now during confinement and what is going on in an egalitarian environment of absolute respect so that they feel free to tell how they feel. The rest of the people usually offer advice and assistance as it is a space in which we promote solidarity". (TP3)

The principal of the PS3 school highlights that the principles on which DG are based allow us to recover the feeling of a group despite confinement and physical isolation (PP2).

Among the DGs, we find dialogic literary gatherings (DLGs) where the best literary works created by humanity are discussed. These are the groups on which the CFG participants offered the most reflections. The principal of the PS1 school, who is also the moderator of the DLG in Grade 2, explains that they are reading an adapted version for children of *The Odyssey* by Homer. She states that the interventions that the children make regarding this work are usually related to violence and gender violence; in dialogic literary gatherings, it is common for discussions about violence and gender violence to emerge (PP1). She also explains that, starting with the interventions of the children, debates emerge about the importance of friendship and of contacting friends, especially at times such as the confinement they are currently experiencing.

The school counsellor from the SS2 secondary school explained that at the "1st of Bachillerato" (the equivalent of Grade 11 in the US and Year 11 in the UK), they had a DLG on *1984* by George Orwell. In the discussion of the text, students constantly connected ideas in the text to the confinement situation and promoting the importance of supportive relationships to better handle the situation. She said that the DLG has revealed the positive aspects that this situation can teach and offer about the need for relationships and solidarity among people (CS1). The DLG opened doors in the homes of the students during confinement days. The teacher of the SS1 secondary school explains that it has had a positive impact on all students, but particularly on students who do not have other spaces where they can share their feelings about the situation in which they are living: they need a space to share experiences with the rest of the students that especially favours the most vulnerable who, on many occasions, do not have other spaces to share (TS1). The principal from the ES3 special education school argues that the DLG that they have conducted in their school has enabled it to be a space of interaction and learning (PE1), which is necessary for all children, but especially for the most vulnerable ones.

### 3.3. Class Assembly or Mentoring

Some of these schools organise weekly class assemblies or mentoring. Some maintained these assemblies online during the period of confinement in which we have developed the fieldwork. Some schools call these assemblies and others call them mentoring, but the aim is the same. These spaces are usually led by the teacher who is responsible for a particular class group to work with the students on topics related to social cohesion, the prevention of abuse and violence, and promoting dialogue among the students. Some teachers noted the importance of maintaining the online assemblies in the situation of confinement as another space to "open doors" and a way for the students to share how they are living their relationships these days. The principal of PS3 school states,

> "It is important that the assemblies go on in order to report violence, to allow them to express how they are living these days, about the relations of friendship and maybe to have a space to report violence". (PP2)

The principal of PS1 considers the assemblies a space that contributes to creating a safe environment because the students recall the evidence-based messages that they have worked on for the promotion

of zero violence in their relations. She perceives the assembly as a protective environment, preventive, which keeps alive the international orientations to prevent and stop violence (PP1).

The teacher of SS1 explains that in her school, they were decided to continuing with the assemblies during the confinement period because the students spent more time connected to social networks and were more exposed to different types of abuse. She explains that teachers and students have observed an increase in offensive messages online (TS1), and they have been able to talk about this in the assemblies.

### 3.4. Dialogic Pedagogical Gatherings with Teachers and Community

All the teachers who participated in the study participated in dialogic pedagogical gatherings (DPGs) before confinement.

In DPGs, teachers jointly read relevant books about theories about learning from authors such as Vygotsky, Freire, and Bruner, as well as articles published in scientific impact journals with a great variety of topics related to education [36]. Among the topics are papers about school-based programmes to prevent child abuse and violence. Participants discuss these texts in connection with their daily educational practices. This is a very enriching activity that allows teachers to reflect upon the purpose of their practice and gives them skills to act in the classroom according to the best educational theories and the most recent scientific contributions. The principal of the ES3 special education school expresses the importance of keeping this space for training during confinement to be able to give better answers to the complex situations of their students:

> "It is an action that helps us learn and transform our educational practice. It is a very important moment because we share learning, we meet and we reflect upon how to improve our actions in these moments. We have now read the chapters about the prevention of violence and friendship of one of the guides from the Child Study Center of Yale University. We have been able to think about the importance of doing [these actions] very well if situations of violence arise among our students in their families during confinement. We have also talked about the importance of maintaining relationships of friendship between children because this is a constraint in their lives, how to keep promoting healthy relationships, helping them to choose their friends well, continuing to accompany them in these moments". (PE1)

The PS1 school has continued to conduct a DGP only with teachers and has created another one that is open to families and the community. This second DPG is specific to articles on violence prevention and child abuse. From this school, contacts have been established with the Equality Committee of the City Council so that this online DPG can be open for participation by all the schools in the city. The director of PS1 explains how this gathering helps teachers, families and other members of the community identify the elements in their daily interactions that can prevent violence and abuse, thereby contributing to creating a safe environment for children.

### 3.5. Mixed Committees and Community Networks

In some of the participating schools, the families and the community directly collaborate with the teachers in the organisation of the school through mixed committees. Some of these mixed committees, which are specifically in charge of promoting actions for the prevention of violence and different types of child abuse, already existed before confinement.

There were four main strategies developed by the mixed committees. First, making sure that contact was established with all school families and all students through telephone calls, Telegram groups, and peer-to-peer communication. Second, exchanging information on economic aid and food scholarships. Third, making available particular resources from the school to the families, such as the distribution of electronic devices and the lending of school supplies to the families most in need, through a contractual agreement with them. Lastly, ensuring that children have access to supportive relationships in safe contexts free of violence.

In particular, PS1 has created a network of solidarity that includes the families and the community. A group of approximately 13 people from different families are helping to contact families that may have problems with their children and to connect them to the different activities of the school. In this group, families and teachers are looking for solutions for each of the families that have difficulties connecting. To provide solutions, this group is in contact with other institutions of the community, such as the Department of Education and the Department of Equality of the City Council.

In PS2, a group of families, teachers, lunch monitors and volunteers are in charge of promoting a sense of community to manage isolation. They do so through the dynamisation of social networks with positive messages of solidarity, and organising online activities such as debates and film clubs (where films are discussed). The teacher in this school explains that through this group they want to promote supportive relationships and friendship: "the creation of this solidarity group has the aim of weaving a type of solidarity from which a friendship stems" (TP1).

*3.6. Dynamisation of Social Networks with Preventive Messages and the Creation of a Sense of Community*

All the schools that participated in the study know very well that communication through social networks contributes to "open doors" and promotes a sense of community despite confinement. Therefore, all these schools actively communicate through social networks such as Facebook, Twitter, WhatsApp, and Telegram. In these communications, messages promote supportive relationships and a sense of community; for example, "We are connected! If you need anything, we can help you!"; "Are you overwhelmed by the situation? Do you need us to help you out? What is worrying you? We are here♡"; "Friendships are our lighthouse in this journey"; #Wearenotalone; #ConnectedFromHome; "Friendships are similar to the stars, they shine brighter in the dark"; "Love is strong and brave"; #ConfinementWithLove"; and "Can we help one person every day with a sign of love? Are you in?" In addition, they are disseminating messages through social networks for the prevention of child abuse and violence, such as, "In confinement 0 violence; we are upstanders!" and "We act and we take a stand".

These messages on social networks are one more action that contributes to creating a climate where positioning against violence and abuse is still present during confinement. Standing up in this way is what is known as "bystander intervention" which has been successful in reducing violence and abuse in the various social and educational settings in which it has been implemented [37]. The fact that these schools show in their social networks that every person in the educational community is an active bystander who can stop or report a situation of abuse is fundamental to creating a protective and trusting climate, both for children who may suffer abuse and for those who report it.

## 4. Discussion

Six actions have been implemented with the goal of "opening doors" to foster supportive relationships and a safe environment to prevent child abuse during COVID-19 confinement: dialogic workspaces, dialogic gatherings with students, class assemblies or mentoring, dialogic pedagogical gatherings with teachers and community, mixed committees and community networks, and social network dynamisation with preventive messages and the creation of a sense of community.

These actions include elements identified by previous research on effective school-based programmes to prevent child abuse and encourage the preventive socialisation of violence. In most of the participating schools, specific actions are not implemented with the sole aim of increasing children's knowledge of child abuse concepts and self-protection skills, as in other programmes [6,15,16]; rather, these schools promote a safe environment with regard to all interactions that involve teachers, students, families, and other community members [26,28,48]. This approach was already being used by these schools before confinement. After confinement began, they adapted these actions to the new situation.

In the online dialogic workspaces created in four of the participating schools, the aim was not only to prevent students from being left behind in learning mathematics or language. Another key objective of this action has been to "open doors" to break physical isolation by maintaining supportive relationships [12]. Therefore, a space for work and dialogue has been created in the homes, where the

children are connected to their peers and a trusted adult (teacher or volunteer). This is a space where the children feel safe to share how they are experiencing confinement [6,17].

Creating trusted environments involves not only small online workspaces but also their extension to the community by engaging families [18], fostering the responsibility of society to ensure the safety of children [15]. Five of the schools have promoted solidarity networks through mixed committees formed by teachers, families and other community members [19]. These committees look for solutions in the community to ensure that all children can access resources to be able to connect. They also attempt to ensure that at least one adult reference person, whether a teacher or a volunteer, contacts families who may be having more difficulties. Teachers also make daily calls to students who may be at risk of abuse or neglect [13], and they are in constant contact with other social and educational services in the city.

The schools are very active on social networks, such as Facebook, Twitter, and WhatsApp, which contributes to a sense of community [33]. The principals or teachers are community managers. Social networks strengthen the sense of community in the face of isolation with messages such as, "Are you overwhelmed by the situation? Do you need help? What are you worried about? We are here". They also post messages that contribute to community involvement to build safe environments against child abuse and violence, such as, "In confinement 0 Violence. We are upstanders! We act and we take a stand!"

Class assemblies are among the actions that many of these schools were taking prior to confinement. Five of the schools have maintained these activities during confinement. The assembly is perhaps most specifically where children's knowledge and skills to prevent abuse and violence are directly addressed [22–24]. At the same time, the assembly is another space for dialogue where students share their daily experiences and concerns, which is one of the elements identified in research on effective programmes to prevent child abuse [32]. The class assembly is one of the spaces where students recall and reflect on the key messages of abuse and violence prevention from their own experiences, such as the importance of being an upstander by taking a stand for the victim and stopping and reporting abusive behaviour [37–39]. As some of the participating teachers have noted, the assembly is a space for zero violence where they can report situations of abuse that they have witnessed or experienced in their relationships between peers during confinement [27]. In this sense, although teachers cannot control the contexts of the children participating virtually from home, what the evidence tells us is that the more spaces for dialogue and positive relationships children have, the more opportunities exist to prevent and overcome child abuse [10,11].

As indicated previously, prior to confinement, all the schools in the study applied the evidence-based actions to prevent violence and abuse, from which the ODAs are recreated. Therefore, all teachers had participated in dialogic teacher training processes [51] for at least two years and have been trained in evidence-based actions for the prevention of violence.

The active involvement of students through interactive activities is another element identified in effective school-based programmes to prevent abuse [6,22,28,31]. The dialogic gatherings are one of the actions that these schools had already implemented and that some have maintained. Specifically, dialogic literary gatherings are an action that was already being conducted in these schools, and seven of them have continued to do so online during confinement. In the DLGs, through the reading and joint discussion of the best literary creations of humanity, a type of dialogue is promoted among the students that leads them to compare these works to their own experiences [52]. Research on effective programmes to prevent child abuse highlights the use of stories in which students can identify themselves, share their views, and listen to the opinions of others, thereby increasing their skills in identifying possible real-life abuse situations [22]. Teachers explain how DLGs on *The Odyssey* promote in-depth discussions about violence or friendship among primary school students. A high school teacher explained how students who read and discussed the book *1984* by George Orwell reflected on the need for supportive relationships to overcome difficult situations due to the pandemic and confinement. A feeling of friendship is promoted in the DLGs, which is identified as one of the protective factors against abuse [29].

Research analysing teacher training in child abuse prevention programmes has highlighted the effectiveness of programmes aimed at developing teachers' potential as agents of social change in their daily work at schools [34,35]. Dialogic pedagogical gatherings (DPGs) with teachers are one of the actions that promote teachers as agents of social change for the prevention of child abuse [26]. Some teachers from the participating schools engaged in these types of gatherings before confinement [36]. Two of the participating schools conducted online DPGs during lockdown. In these gatherings, teachers, along with other community agents, read and discuss relevant books and articles based on scientific evidence on violence and child abuse prevention, which helps them to be active agents in the promotion of safe environments where children feel confident and believed when reporting situations of abuse [17]. In these schools, they have also opened the DPGs to family and community participation. They therefore continue to involve the whole community in the creation of safe environments for children, which is also one of the elements identified in research on effective programmes to prevent child abuse [6,31].

The present study was conducted through intense work by both the researchers and the teachers of the schools involved, with the aim of making it available as soon as possible. In this way, evidence-based actions to "open doors" can be offered and eventually be useful to school teams that would like to promote supportive relationships and safe environments during confinement. Future research directions could collect impact evidence from these actions, including the stories of the teachers, families, and students involved.

## 5. Expected Impact of Open Doors Actions

The COVID-19 crisis has surprised us without an effective response to the increase in child abuse [53]. This is a pending challenge for researchers, social activists, politicians and international organizations in defense of the children. The knowledge generated about ODAs can be useful today and for future crises, placing the school as an active agent in preventing child abuse. In this sense, and even knowing the limitations of the present study due to it being in its initial stage, there are some learnings that may already be useful for schools and public policies.

The main implication of ODAs is to transfer evidence-based interventions for violence and abuse prevention to contexts of confinement. ODAs enable the creation of learning and coexistence contexts between equals and other community agents based on good treatment. This prevents children from experiencing an increase in violence in virtual communication and learning spaces. These actions create, in those schools that apply them, an environment where the interactions valued are based on values and feelings such as friendship, love, kindness, and solidarity. Through them, a sense of community is created that is protective and preventive of child abuse, which enables more people to reject violence and protect children [54] One of the limitations to take into account in the face of the transfer of ODAs is technological availability and access to the Internet of the students.

The main implication of ODAs to public policies is providing evidence-based actions to improve the effectiveness of the programmes in childcare in the days of COVID-19. ODAs enable the creation of evidence-based school plans to contribute to child abuse prevention while promoting learning and a sense of school community in periods of confinement. Preventing the negative impact on children of confinement due to the COVID-19 crisis is not only the responsibility of families and schools but also that of the governments [55]. International research suggests that child abuse prevention and intervention programmes should be based on scientific evidence [56]. One of the limitations of this study, on which work continues, is the incorporation of policymakers in the transfer of ODAs as a public policy [57]. The transfer of ODAs to public policies would contribute to preparing the new educational normality more in line with scientific and ethical criteria for children.

## 6. Conclusions

The actions that are presented with the goal of "opening doors" to promote supportive relationships and safe environments to prevent child abuse during confinement are only some of the many actions

that the educational community has implemented since the closing of the schools. This study is the result of the inclusion of elements that research on child abuse and violence prevention has indicated as effective, as well as dialogue and in-depth reflection with the teachers who are implementing them.

These are not the only actions that can be implemented by schools for the promotion of supportive relationships and for the creation of safe contexts of interaction to "open doors" to physical isolation in homes. However, these actions can be an example for those who want to boost evidence-based actions with this aim. The real promotion of the sense of community, the creation of safe environments where the entire community acts as upstanders, stopping violence and taking a stand for the victim, the creation of spaces where quality learning occurs at the same time as dialogues that give meaning to the topics that are the children's concerns, creating an environment of confidence where they feel heard and supported—all of these are fundamental elements to be considered. Many teachers, families, and communities are agents of social change that create supportive relationships and safe environments that protect childhood.

**Author Contributions:** Conceptualization, E.R., P.M. and M.A.P.-R.; Funding acquisition, R.G.-C.; Investigation, E.R., P.M. and M.A.P.-R.; Methodology, E.R., P.M. and M.A.P.-R.; Visualization, R.G.-C.; Writing—original draft, E.R.; Writing—review & editing, R.G.-C. All authors have read and agreed to the published version of the manuscript.

**Funding:** This research received no external funding.

**Conflicts of Interest:** The authors declare no conflict of interest.

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
