# Peer review of "Schools That ‘Open Doors’ to Prevent Child Abuse in Confinement by COVID-19"

_sustainability, doi:10.3390/su12114685_

Round 1

Reviewer 1 Report

The article object of this review meets the scientific criteria for its publication.

Below are four recommendations for improvement and subsequent publication:

1.

1.3. School-based programs to prevent child abuse

163. The objective of the article should be further specified, that is, that the article seeks to describe evidence-based actions.

The article describes 6 very interesting and scientifically based actions. However, it would be interesting to leave a greater record that the article does not show results resulting from the application of said actions, although if the results of these actions are said to be the subject of other studies and articles that are in process, it would be good to explain it more. and substantiate it.

2.

Throughout the article, it should be explained more that the objective is to open the doors with those actions that are described to avoid abuse in children, however, there would be a greater explanation that these actions are not intended to identify the victims if not that these actions can help prevent.

4.

3.7. Add a paragraph where it is more explained how this action helps to prevent abuse in children, it is explained a little superficial.

5.

2.1. School selection criteria A greater explanation and justification of the choice of the sample would be needed (apart from the fact that it was for convenience and the centers that are applying these actions), types of centers, type of students, ..

Author Response

Estimado crítico,

Especifico las revisiones que se han realizado, indicando el número de línea y la modificación exacta en el documento adjunto.

También encontrará adjunto el archivo del artículo en "Seguimiento de cambios".

Saludos cordiales,

Esther Roca

Reviewer 2 Report

The paper is useful not only for the research community but for the schools and the whole society. I can only congratulate the authors, a congratulation that must be extended to the schools that are fighting child abuse.

I have only one suggestion (double sided) for the authors:

The section devoted to Discussion offers no option to discussion; from my point of view is a (perfect) collection of results. Maybe this can be improved offering some opportunity to this purpose.

On the other hand, this section could be complemented with a subsection of Implications for schools that are implementing these kind of programmes, and for those that are not; and Implications for principals, teachers, counsellors and even the Department of Education of each Autonomous Community. Everyone needs to learn from good practices.

Author Response

Dear reviewer,

I specify the revisions that have been made, indicating the line number and the exact modification in the attached document.

Please find attached the article file in "Track Changes", too.

Kind regards,

Esther Roca

Reviewer 3 Report

There are significant issues around how to protect children who are not attending school due to government restrictions related to Covid-19. As such this paper makes an important contribution in terms of suggesting ways in which schools can continue to play an appropriate safeguarding role. I just have some minor comments:

The statistics from the ANAR foundation are useful in terms of providing a context but I wonder whether it would be possible to present a comparison of statistics pre and post Covid-19 in order to provide a better overview of the impact the pandemic has had. 

The methods are appropriate especially considering the tight timescales but I wondered whether approval was sought from an ethics committee. Secondly 9 schools participated but there were only interviews with staff from 5 of these schools. Were all 9 schools invited to undertake an interview and if not how did the researchers decide which schools to include? 

Finally I wondered whether there should be more discussion about the potential constraints to delivering some of these kinds of sessions. For example, the researchers talk about things like assemblies being a safe place for children to share their experiences of confinement including abuse and violence they have witnessed or experienced but how can teachers  determine whether children are in fact in a safe space in the house away from parents/siblings listening. Otherwise I am concerned that these kinds of sessions, while well intentioned, might actually place children at higher risk. 

Finally the paper is for the most part well written but there are a small number of typos. 

Author Response

(The authors gave the same response as above.)
